# FINDING EQUILIBRIA IN BILINEAR ZERO-SUM GAMES VIA A CONVEXITY-BASED APPROACH

## ABSTRACT

We focus on the design of algorithms for finding equilibria in 2-player zero-sum games. Although it is well known that such problems can be solved by a single linear program, there has been a surge of interest in recent years, for simpler algorithms, motivated in part by applications in machine learning. Our work proposes such a method, inspired by the observation that the duality gap (a standard metric for evaluating convergence in general min-max optimization problems) is a convex function for the case of bilinear zero-sum games. To this end, we analyze a descent-based approach, variants of which have also been used as a subroutine in a series of algorithms for approximating Nash equilibria in general non-zero-sum games. In particular, we analyze a steepest descent approach, by finding the direction that minimises the directional derivative of the duality gap function and move towards that. Our main theoretical result is that the derived algorithms achieve a geometric decrease in the duality gap and improved complexity bounds until we reach an approximate equilibrium. Finally, we complement this with an experimental evaluation. Our findings reveal that for some classes of zero-sum games, the running time of our method is comparable with standard LP solvers, even with thousands of available strategies per player.

## 1 INTRODUCTION

Our work focuses on the design of algorithms for finding Nash equilibria in 2-player bilinear zero-sum games. Zero-sum games have played a fundamental role both in game theory, being among the first classes of games formally studied, and in optimization, as it is easily seen that their equilibrium solutions correspond to solving a min-max optimization problem. It has also been long known that a max-min strategy (i.e., an equilibrium strategy) for each player can can be actually tackled via linear programming. Even further, solving zero-sum games is in fact equivalent to solving linear programs, as properly demonstrated in Adler (2013).

Despite these positive news and the fact that a single linear program (and its dual) suffices to find a Nash equilibrium, there has been a surge of interest in recent years, for simpler algorithms, motivated in part by applications in machine learning. One reason for this is the fact that we may have very large games to solve, corresponding to LPs with thousands of variables and constraints. A second reason could be that e.g., in learning environments, the players may be using iterative algorithms that can only observe limited information, hence it would be impossible to run a single LP for the entire game. As an additional motivation, finding new algorithms for such a fundamental problem can provide insights that could be of further value and interest.

In parallel to the above considerations, there has also been a stream of works on finding approximate Nash equilibria in general non-zero-sum 2 player games. This is a more difficult problem compared to linear programming, and the efforts so far have led to a variety of algorithms, culminating to the constant factor approximation guarantees achieved in Tsaknakis & Spirakis (2008) and Deligkas et al. (2023). What is interesting about these algorithms is that they start with a descent-based approach by computing the directional derivative of the maximum regret of the two players (i.e., the maximum gain a player can have by deviating). In non-zero-sum games, this does not guarantee that one can reach an approximate equilibrium, with always a small approximation, and therefore the descent method has to be combined with an additional subroutine in order to come up with a desired strategy for each player.

The approaches mentioned above naturally give rise to the question of how would they perform if applied to zero-sum games with the duality gap as the objective function. More specifically, we are interested in evaluating just the descent-based part of such approaches. An indication that this may be promising for zero-sum games is the observation that the duality gap is a convex function for zero-sum games. The duality gap function is simply the sum of the regrets of the two players in a given strategy profile. Therefore, to obtain a $\delta$-approximate equilibrium, it suffices that we find a strategy profile where the duality gap is at most $\delta$.

### 1.1 OUR CONTRIBUTIONS

We analyze a method for finding $\delta$-approximate Nash equilibria in zero-sum games, for any constant $\delta$. The method is essentially an adaptation of the algorithms in Tsaknakis & Spirakis (2008); Deligkas et al. (2023; 2017) which are for general games, tailored to the case of zero-sum games and having the duality gap as the objective function. Inspired by the convexity of the duality gap function, the method is applying a steepest descent approach, where we find in each step the direction that minimises the directional derivative of the duality gap function and move towards that. Our main theoretical result is that the derived algorithm achieves a geometric decrease in the duality gap until we reach an approximate equilibrium. This implies that the algorithm terminates after at most $O\left(\frac{1}{\rho} \cdot \log\left(\frac{1}{\delta}\right)\right)$ iterations with a $\delta$-approximate equilibrium and $\rho$ a fixed constant parameter of our algorithm. Finally, we complement this with an experimental evaluation. Even though the method does need to solve a linear program in each iteration, this turns out to be of much smaller size on average (in terms of the number of constraints) than solving the linear program of the entire game. As a result, our findings reveal that for some classes of zero-sum games the running time of our method is comparable with standard LP solvers, even with thousands of available strategies per player. We therefore conclude that the overall approach deserves further exploration, as there are also potential ways of accelerating its running time, discussed in Section 6.

### 1.2 RELATED WORK

As already mentioned, conceptually, the works most related to ours are Tsaknakis & Spirakis (2008); Deligkas et al. (2023; 2017). Although these papers do not consider zero-sum games, they do utilize a descent-based part as a starting point. The main differences with our work is that first of all, their descent is performed with respect to the maximum regret among the two players, whereas we use the duality gap function. Furthermore the descent phase of their algorithms does not suffice for more general games, and hence their focus is less on the decent phase itself and more on utilizing further procedures to produce approximate equilibria.

There is a plethora of algorithms for linear programming and zero-sum games, which is impossible to list here, but we comment on what we feel are most relevant. When focusing on optimization algorithms for zero-sum games, Hoda et al. (2010) use Nesterov's first order smoothing techniques to achieve an $\epsilon$-equilibrium in $O(1/\epsilon)$ iterations, with added benefits of simplicity and rather low computational cost per iteration. Following up on that work, Gilpin et al. (2012) propose an iterated version of Nesterov's smoothing technique, which runs within $O(\frac{||A||}{\delta(A)} \cdot \ln(1/\epsilon))$ iterations. However, while this is a significant improvement, the complexity depends on a condition measure $\delta(A)$, with $A$ being the payoff matrix, not necessarily bounded by a constant.

Apart from the optimization viewpoint, there has been great interest in designing learning algorithms for zero-sum games, given their relevance in formulating GANs in deep learning Goodfellow et al. (2014) and also other applications in machine learning. Some of the earlier and standard results in this area concern convergence *on average*. I.e., it has been known that by using no-regret algorithms, such as the Multiplicative Weights Update (MWU) methods Arora et al. (2012) the empirical average of the players' strategies over time converges to a Nash equilibrium in zero-sum games. One could also utilize the so-called Gradient Descent/Ascent (GDA) algorithms (which differ from ours since in GDA each person is applying gradient descent to her utility function, which is not convex). In the recent years, there has also been a great interest in the more robust notion of *last-iterate convergence*. This means that the strategy profile $(x_t, y_t)$, reached at iteration $t$ of an iterative algorithm, converges to the actual equilibrium as $t \to \infty$. In fact, it was shown in Bailey & Piliouras (2018) that several MWU variants do not satisfy last-iterate convergence. Motivated by this, there has been a series of

works in the more recent years that has been studying last iterate convergence. The positive results that have been obtained for zero-sum games is that improved variants of Gradient Descent such as the Extra Gradient method Korpelevich (1976) or the Optimistic Gradient method Popov (1980) attain last iterate convergence. The currently best rate is $O\left(1/\sqrt{t}\right)$ in terms of the duality gap Cai et al. (2022), but it is conceivable that better rates could be achieved.

## 2 PRELIMINARIES

We consider bilinear zero-sum games $(\boldsymbol{R}, -\boldsymbol{R})$, with $\boldsymbol{R} \in [0,1]^{n \times n}$ being the payoff matrix of the row player, without loss of generality[1], with $n$ pure strategies for any player. We consider mixed strategies $\boldsymbol{x} \in \Delta^{n-1}$ as a probability distribution (column vector) on the pure strategies of a player, with $\Delta^{n-1}$ be the $(n-1)$-dimensional simplex. We also denote by $\boldsymbol{e}_i$ the distribution corresponding to a pure strategy $i$, with 1 in the index $i$ and zero elsewhere. A strategy profile is a pair $(\boldsymbol{x}, \boldsymbol{y})$, where the row player plays the strategy $\boldsymbol{x}$ and the column player plays the strategy $\boldsymbol{y}$. Under a strategy profile $(\boldsymbol{x}, \boldsymbol{y})$, the expected payoff of the row player is $\boldsymbol{x}^\top \boldsymbol{R} \boldsymbol{y}$ and the expected payoff of the column player is $-\boldsymbol{x}^\top \boldsymbol{R} \boldsymbol{y}$.

A pure strategy $i$ is a $\rho$-best-response strategy against $\boldsymbol{y}$ for the row player, for $\rho \in [0,1]$, if and only if, $\boldsymbol{e}_i^\top \boldsymbol{R} \boldsymbol{y} + \rho \geq \boldsymbol{e}_j^\top \boldsymbol{R} \boldsymbol{y}$, for any $j$. Similarly, a pure strategy $j$ for the column player is a $\rho$-best-response strategy against some strategy $\boldsymbol{x}$ of the row player if and only if $\boldsymbol{x}^T \boldsymbol{R} \boldsymbol{e}_j \leq \boldsymbol{x}^T \boldsymbol{R} \boldsymbol{e}_i + \rho$, for any $i$. Having these, we define as $BR_r^\rho(\boldsymbol{y})$ the set of the $\rho$-best-response pure strategies of the row player against $\boldsymbol{y}$ and as $BR_c^\rho(\boldsymbol{x})$ the set of the $\rho$-best-response pure strategies of the column player against $\boldsymbol{x}$. We now give some of the basic definitions from game theory for zero-sum bimatrix games.

**Definition 1** (Nash equilibrium Nash (1951); Von Neumann (1928))**.** *A strategy profile* $(\boldsymbol{x}^*, \boldsymbol{y}^*)$ *is a Nash equilibrium in the game* $(\boldsymbol{R}, -\boldsymbol{R})$*, if and only if, for any* $i, j$,

$$v = \boldsymbol{x}^{*\top} \boldsymbol{R} \boldsymbol{y}^* \geq \boldsymbol{e}_i^\top \boldsymbol{R} \boldsymbol{y}^*, \text{ and, } v = \boldsymbol{x}^{*\top} \boldsymbol{R} \boldsymbol{y}^* \leq \boldsymbol{x}^{*\top} \boldsymbol{R} \boldsymbol{e}_j,$$

where $v$ is the value of the row player (value of the game).

**Definition 2** ($\delta$-Nash equilibrium)**.** *A strategy profile* $(\boldsymbol{x}, \boldsymbol{y})$ *is a* $\delta$-*Nash equilibrium in the game* $(\boldsymbol{R}, -\boldsymbol{R})$*, with* $\boldsymbol{R} \in [0,1]^{n \times n}$*, and with* $\delta \in [0,1]$*, if and only if, for any* $i, j$,

$$\boldsymbol{x}^\top \boldsymbol{R} \boldsymbol{y} + \delta \geq \boldsymbol{e}_i^\top \boldsymbol{R} \boldsymbol{y}, \text{ and, } \boldsymbol{x}^\top \boldsymbol{R} \boldsymbol{y} - \delta \leq \boldsymbol{x}^\top \boldsymbol{R} \boldsymbol{e}_j.$$

With these at hand, we can now define the regret functions of the players as follows.

**Definition 3** (Regret of a player)**.** *Let* $\boldsymbol{R} \in [0,1]^{n \times n}$ *be the payoff matrix of the row player. Then the regret function* $f_{\boldsymbol{R}} : \Delta^{n-1} \times \Delta^{n-1} \to [0,1]$ *of the row player for the strategy profile* $(\boldsymbol{x}, \boldsymbol{y})$ *is*

$$f_{\boldsymbol{R}}(\boldsymbol{x}, \boldsymbol{y}) = \max_i \boldsymbol{e}_i^\top \boldsymbol{R} \boldsymbol{y} - \boldsymbol{x}^\top \boldsymbol{R} \boldsymbol{y}.$$

*Similarly, for the column player the regret function is*

$$f_{-\boldsymbol{R}}(\boldsymbol{x}, \boldsymbol{y}) = \max_j \boldsymbol{x}^\top (-\boldsymbol{R}) \boldsymbol{e}_j + \boldsymbol{x}^\top \boldsymbol{R} \boldsymbol{y} = -\min_j \boldsymbol{x}^\top \boldsymbol{R} \boldsymbol{e}_j + \boldsymbol{x}^\top \boldsymbol{R} \boldsymbol{y}.$$

An important quantity for evaluating the performance or convergence of algorithms is the sum of the regrets, i.e., the function $V(\boldsymbol{x}, \boldsymbol{y}) = f_{\boldsymbol{R}}(\boldsymbol{x}, \boldsymbol{y}) + f_{-\boldsymbol{R}}(\boldsymbol{x}, \boldsymbol{y}) = \max_i \boldsymbol{e}_i^\top \boldsymbol{R} \boldsymbol{y} - \min_j \boldsymbol{x}^\top \boldsymbol{R} \boldsymbol{e}_j$. This is referred to in the bibliography as the *duality gap* in the case of zero-sum games.

## 3 WARMUP: DUALITY GAP PROPERTIES

We have the following Theorem in the case of zero-sum games for the *duality gap* function $V(\boldsymbol{x}, \boldsymbol{y})$.

---

[1]We can easily see that we can do scaling for any $\boldsymbol{R} \in \mathbb{R}^{n \times n}$ s.t. $\boldsymbol{R} \in [0,1]^{n \times n}$ keeping exactly the same Nash equilibria.

**Theorem 1.** *The duality gap $V(\boldsymbol{x}, \boldsymbol{y})$ is convex in its domain.*

*Proof.* Let $(\boldsymbol{x}_1, \boldsymbol{y}_1)$ and $(\boldsymbol{x}_2, \boldsymbol{y}_2)$ be two arbitrary different strategy profiles, $p \in (0, 1)$ and $(\boldsymbol{x}, \boldsymbol{y}) = p \cdot (\boldsymbol{x}_1, \boldsymbol{y}_1) + (1 - p) \cdot (\boldsymbol{x}_2, \boldsymbol{y}_2) = (p \cdot \boldsymbol{x}_1 + (1 - p) \cdot \boldsymbol{x}_2, p \cdot \boldsymbol{y}_1 + (1 - p) \cdot \boldsymbol{y}_2)$ be a convex combination of them. Then, we have

$$
\begin{aligned}
V(\boldsymbol{x}, \boldsymbol{y}) &= V(p \cdot \boldsymbol{x}_1 + (1 - p) \cdot \boldsymbol{x}_2, p \cdot \boldsymbol{y}_1 + (1 - p) \cdot \boldsymbol{y}_2) \\
&= \max_i \boldsymbol{e}_i^\top \boldsymbol{R}(p \cdot \boldsymbol{y}_1 + (1 - p) \cdot \boldsymbol{y}_2) - \min_j (p \cdot \boldsymbol{x}_1 + (1 - p) \cdot \boldsymbol{x}_2)^\top \boldsymbol{R} \boldsymbol{e}_j \\
&\leq p \cdot \max_i \boldsymbol{e}_i^\top \boldsymbol{R} \boldsymbol{y}_1 + (1 - p) \cdot \max_i \boldsymbol{e}_i^\top \boldsymbol{R} \boldsymbol{y}_2 \\
&\quad - p \cdot \min_j \boldsymbol{x}_1^\top \boldsymbol{R} \boldsymbol{e}_j - (1 - p) \cdot \min_j \boldsymbol{x}_2^\top \boldsymbol{R} \boldsymbol{e}_j \\
&= p \cdot \max_i \boldsymbol{e}_i^\top \boldsymbol{R} \boldsymbol{y}_1 + (1 - p) \cdot \max_i \boldsymbol{e}_i^\top \boldsymbol{R} \boldsymbol{y}_2 - p \cdot \min_j \boldsymbol{x}_1^\top \boldsymbol{R} \boldsymbol{e}_j - (1 - p) \cdot \min_j \boldsymbol{x}_2^\top \boldsymbol{R} \boldsymbol{e}_j \\
&= p \cdot V(\boldsymbol{x}_1, \boldsymbol{y}_1) + (1 - p) \cdot V(\boldsymbol{x}_2, \boldsymbol{y}_2),
\end{aligned}
$$

the first inequality holds by the convexity and concavity of the $\max$ and the $\min$ function, respectively. $\qquad\square$

We now give a connection of the *duality gap* and the Nash equilibria.

**Theorem 2.** *A strategy profile $(\boldsymbol{x}^*, \boldsymbol{y}^*)$ is a Nash equilibrium of the game $(\boldsymbol{R}, -\boldsymbol{R})$, if and only if, is a (global) minimum[2] of the function $V(\boldsymbol{x}, \boldsymbol{y})$.*

*Proof.* Let $(\boldsymbol{x}^*, \boldsymbol{y}^*)$ be a Nash equilibrium, then it holds $V(\boldsymbol{x}^*, \boldsymbol{y}^*) = 0$ by the definition of the NE, but since the values of $V(\boldsymbol{x}, \boldsymbol{y}) \in [0, 2]$ this implies that $(\boldsymbol{x}^*, \boldsymbol{y}^*)$ is a global minimum of the function in its domain. Let now a strategy profile $(\boldsymbol{x}, \boldsymbol{y})$ such that $V(\boldsymbol{x}, \boldsymbol{y}) = 0 = f_{\boldsymbol{R}}(\boldsymbol{x}, \boldsymbol{y}) + f_{-\boldsymbol{R}}(\boldsymbol{x}, \boldsymbol{y})$, this trivially implies that $f_{\boldsymbol{R}}(\boldsymbol{x}, \boldsymbol{y}) = 0$ and $f_{-\boldsymbol{R}}(\boldsymbol{x}, \boldsymbol{y}) = 0$ since $f_{\boldsymbol{R}}, f_{-\boldsymbol{R}} \in [0, 1]$, thus we have that $(\boldsymbol{x}, \boldsymbol{y})$ is a NE in the zero-sum game. $\qquad\square$

In a very similar manner to the previous theorem, we can also have the following.

**Theorem 3.** *Let $(\boldsymbol{x}, \boldsymbol{y})$ be a strategy profile in a zero-sum game. If $V(\boldsymbol{x}, \boldsymbol{y}) \leq \delta$, then $(\boldsymbol{x}, \boldsymbol{y})$ is a $\delta$-NE.*

## 4 A GRADIENT DESCENT-BASED ALGORITHM FOR ZERO-SUM GAMES

In this section, we present an algorithm based on a gradient-descent approach for the function $V(\boldsymbol{x}, \boldsymbol{y})$ in zero-sum bimatrix games. The algorithm can be seen as an adaptation[3] of the descent phase used in Tsaknakis & Spirakis (2008); Deligkas et al. (2017; 2023) for general games, tailored here for zero-sum games. The main idea behind the algorithm is that since the global minimum of the duality gap function $V(\boldsymbol{x}, \boldsymbol{y})$ is a Nash equilibrium and the duality gap is a convex function for zero-sum bilinear games, we use a descent method based on the directional derivative of the function $V(\boldsymbol{x}, \boldsymbol{y})$. To identify the direction that minimizes the directional derivative at every step we use linear programming (albeit solving much smaller linear programs on average than the program describing the zero-sum game itself).

To start our analysis, we define first the directional derivative.

**Definition 4.** *The directional derivative of the duality gap at a point $(\boldsymbol{x}, \boldsymbol{y})$ with respect to a direction $(\boldsymbol{x}', \boldsymbol{y}') \in \Delta^{n-1} \times \Delta^{n-1}$ is the limit, if it exists,*

$$
\nabla_{(\boldsymbol{x}', \boldsymbol{y}')} V(\boldsymbol{x}, \boldsymbol{y}) = \lim_{\varepsilon \searrow 0} \frac{V\big((1 - \varepsilon) \cdot (\boldsymbol{x}, \boldsymbol{y}) + \varepsilon \cdot (\boldsymbol{x}', \boldsymbol{y}')\big) - V(\boldsymbol{x}, \boldsymbol{y})}{\varepsilon},
$$

---

[2]Note that the set of the Nash equilibria in zero-sum games and the set of the optimal solutions of the *duality gap* are convex and identical to each other.

[3]Here as the objective function we use the sum of the regrets instead of the maximum of the two regrets.

We provide below a more convenient form for the directional derivative that facilitates the remaining analysis.

**Theorem 4.** *The directional derivative of the duality gap $V$ at a point $(\boldsymbol{x}, \boldsymbol{y})$ with respect to a direction $(\boldsymbol{x}', \boldsymbol{y}') \in \Delta^{n-1} \times \Delta^{n-1}$, is given by*

$$\nabla_{(\boldsymbol{x}', \boldsymbol{y}')} V(\boldsymbol{x}, \boldsymbol{y}) = \max_{i \in BR_r(\boldsymbol{y})} \boldsymbol{e}_i^\top \boldsymbol{R} \boldsymbol{y}' - \min_{j \in BR_c(\boldsymbol{x})} (\boldsymbol{x}')^\top \boldsymbol{R} \boldsymbol{e}_j - V(\boldsymbol{x}, \boldsymbol{y}).$$

Furthermore, by the definition of directional derivative we have the following consequence.

**Lemma 1.** *Let $(\boldsymbol{x}, \boldsymbol{y})$ be a strategy profile that is not a $\delta$-Nash equilibrium, for any $\delta \in [0, 1]$, in the game $(\boldsymbol{R}, -\boldsymbol{R})$, then*

$$\nabla_{(\boldsymbol{x}', \boldsymbol{y}')} V(\boldsymbol{x}, \boldsymbol{y}) < -\delta,$$

*where $(\boldsymbol{x}', \boldsymbol{y}') \in \Delta^{n-1} \times \Delta^{n-1}$ is a direction that minimizes the directional derivative.*

*Proof.* If $(\boldsymbol{x}, \boldsymbol{y})$ is not a $\delta$-Nash equilibrium, then it holds that $V(\boldsymbol{x}, \boldsymbol{y}) > \delta$. Assume now that $\nabla_{(\boldsymbol{x}', \boldsymbol{y}')} V(\boldsymbol{x}, \boldsymbol{y}) \geq -\delta$ for the direction $(\boldsymbol{x}', \boldsymbol{y}')$ that minimizes the directional derivative. This implies that $\nabla_{(\hat{\boldsymbol{x}}, \hat{\boldsymbol{y}})} V(\boldsymbol{x}, \boldsymbol{y}) \geq -\delta$ for any direction $(\hat{\boldsymbol{x}}, \hat{\boldsymbol{y}})$. Then, we have that $\max_{i \in BR_r(\boldsymbol{y})} \boldsymbol{e}_i^\top \boldsymbol{R} \hat{\boldsymbol{y}} - \min_{j \in BR_c(\boldsymbol{x})} (\hat{\boldsymbol{x}})^\top \boldsymbol{R} \boldsymbol{e}_j - V(\boldsymbol{x}, \boldsymbol{y}) \geq -\delta$, which implies that $V(\boldsymbol{x}, \boldsymbol{y}) \leq \delta + \max_{i \in BR_r(\boldsymbol{y})} \boldsymbol{R} \hat{\boldsymbol{y}} - \min_{j \in BR_c(\boldsymbol{x})} (\hat{\boldsymbol{x}})^\top \boldsymbol{R}$, for any direction $(\hat{\boldsymbol{x}}, \hat{\boldsymbol{y}})$, thus even for a Nash equilibrium $(\boldsymbol{x}^*, \boldsymbol{y}^*)$. But at an equilibrium $(\boldsymbol{x}^*, \boldsymbol{y}^*)$, we have $\max_{i \in BR_r(\boldsymbol{y})} \boldsymbol{e}_i^\top \boldsymbol{R} \boldsymbol{y}^* - \min_{j \in BR_c(\boldsymbol{x})} (\boldsymbol{x}^*)^\top \boldsymbol{R} \boldsymbol{e}_j \leq v - v = 0$. Thus, $V(\boldsymbol{x}, \boldsymbol{y}) \leq \delta$, which implies that $(\boldsymbol{x}, \boldsymbol{y})$ is a $\delta$-Nash equilibrium, thus a contradiction to our initial assumption. $\square$

In a similar manner to Theorem 4, we define below an approximate version of the directional derivative. The reason we do that will become clear later on, in order to show that the duality gap decreases from one iteration of the algorithm to the next. The main idea in the definition below is to include approximate best responses in the maximization and minimization terms involved in Theorem 4. Namely, for $\rho > 0$, let $BR_r^\rho(\boldsymbol{y})$ be the set of $\rho$-approximate best response strategies of the row player against strategy $\boldsymbol{y}$ of the column player. More formally, $i \in BR_r^\rho(\boldsymbol{y})$, if and only if, $\boldsymbol{e}_i^\top \boldsymbol{R} \boldsymbol{y} + \rho \geq \boldsymbol{e}_j^\top \boldsymbol{R} \boldsymbol{y}$, for any $j$. Similarly, let $BR_c^\rho(\boldsymbol{x})$ be the set of $\rho$-approximate best response strategies of the column player against strategy $\boldsymbol{x}$ of the column player.

**Definition 5** ($\rho$-directional derivative). *The $\rho$-directional derivative of the duality gap $V$ at a point $(\boldsymbol{x}, \boldsymbol{y})$ with respect to a direction $(\boldsymbol{x}', \boldsymbol{y}') \in \Delta^{n-1} \times \Delta^{n-1}$, is given by*

$$\rho\text{-}\nabla_{(\boldsymbol{x}', \boldsymbol{y}')} V(\boldsymbol{x}, \boldsymbol{y}) = \max_{i \in BR_r^\rho(\boldsymbol{y})} \boldsymbol{e}_i^\top \boldsymbol{R} \boldsymbol{y}' - \min_{j \in BR_c^\rho(\boldsymbol{x})} (\boldsymbol{x}')^\top \boldsymbol{R} \boldsymbol{e}_j - V(\boldsymbol{x}, \boldsymbol{y}).$$

**Lemma 2.** *It holds that for any direction $(\boldsymbol{x}', \boldsymbol{y}') \in \Delta^{n-1} \times \Delta^{n-1}$, and for any $\rho > 0$,*

$$\nabla_{(\boldsymbol{x}', \boldsymbol{y}')} V(\boldsymbol{x}, \boldsymbol{y}) \leq \rho\text{-}\nabla_{(\boldsymbol{x}', \boldsymbol{y}')} V(\boldsymbol{x}, \boldsymbol{y}).$$

*Proof.* By definitions, we have that

$$\nabla_{(\boldsymbol{x}', \boldsymbol{y}')} V(\boldsymbol{x}, \boldsymbol{y}) = \max_{i \in BR_r(\boldsymbol{y})} \boldsymbol{e}_i^\top \boldsymbol{R} \boldsymbol{y}' - \min_{j \in BR_c(\boldsymbol{x})} (\boldsymbol{x}')^\top \boldsymbol{R} \boldsymbol{e}_j - V(\boldsymbol{x}, \boldsymbol{y})$$

$$\leq \max_{i \in BR_r^\rho(\boldsymbol{y})} \boldsymbol{e}_i^\top \boldsymbol{R} \boldsymbol{y}' - \min_{j \in BR_c^\rho(\boldsymbol{x})} (\boldsymbol{x}')^\top \boldsymbol{R} \boldsymbol{e}_j - V(\boldsymbol{x}, \boldsymbol{y})$$

$$= \rho\text{-}\nabla_{(\boldsymbol{x}', \boldsymbol{y}')} V(\boldsymbol{x}, \boldsymbol{y}),$$

the first inequality holds since, by definition, $BR_r(\boldsymbol{y}) \subseteq BR_r^\rho(\boldsymbol{y})$ and $BR_c(\boldsymbol{x}) \subseteq BR_c^\rho(\boldsymbol{x})$. $\square$

**Lemma 3.** *Let $(\boldsymbol{x}, \boldsymbol{y})$ be a strategy profile that is not a $\delta$-Nash equilibrium, for any $\delta \in [0, 1]$, in the game $(\boldsymbol{R}, -\boldsymbol{R})$, then*

$$\rho\text{-}\nabla_{(\boldsymbol{x}', \boldsymbol{y}')} V(\boldsymbol{x}, \boldsymbol{y}) < -\delta,$$

*where $(\boldsymbol{x}', \boldsymbol{y}') \in \Delta^{n-1} \times \Delta^{n-1}$ is a direction that minimizes the $\rho$-directional derivative.*

*Proof.* Let assume that

$$\rho \cdot \nabla_{(\boldsymbol{x}',\boldsymbol{y}')} V(\boldsymbol{x},\boldsymbol{y}) \geq -\delta \implies$$

$$\max_{i \in BR_r^\rho(\boldsymbol{y})} \boldsymbol{e}_i^\top \boldsymbol{R}\boldsymbol{y}' - \min_{j \in BR_c^\rho(\boldsymbol{x})} (\boldsymbol{x}')^\top \boldsymbol{R}\boldsymbol{e}_j + \delta \geq V(\boldsymbol{x},\boldsymbol{y}) \implies$$

$$\delta \geq V(\boldsymbol{x},\boldsymbol{y})$$

since $\max_{i \in BR_r^\rho(\boldsymbol{y})} \boldsymbol{e}_i^\top \boldsymbol{R}\boldsymbol{y}' - \min_{j \in BR_c^\rho(\boldsymbol{x})} (\boldsymbol{x}')^\top \boldsymbol{R}\boldsymbol{e}_j \leq 0$, since for the NE $(\boldsymbol{x}^*, \boldsymbol{y}^*)$ it holds that $\max_{i \in BR_r^\rho(\boldsymbol{y})} \boldsymbol{e}_i^\top \boldsymbol{R}\boldsymbol{y}^* - \min_{j \in BR_c^\rho(\boldsymbol{x})} (\boldsymbol{x}^*)^\top \boldsymbol{R}\boldsymbol{e}_j \leq v - v = 0$. $\qquad\square$

## 4.1 THE ALGORITHM

We now present our algorithm.

---
**Algorithm 1: The gradient descent-based algorithm.**

INPUT: A zero-sum game $(\boldsymbol{R}, -\boldsymbol{R})$, an approximation parameter $\delta \in (0, 1]$, a constant $\rho \in (0, 1]$, and a constant $\epsilon \in (0, 1]$.
OUTPUT: A $\delta$-NE strategy profile.

Pick an arbitrary strategy profile $(\boldsymbol{x}, \boldsymbol{y})$.
While $V(\boldsymbol{x}, \boldsymbol{y}) > \delta$
  $(\boldsymbol{x}', \boldsymbol{y}')$ = FIND_DIRECTION$(\boldsymbol{x}, \boldsymbol{y}, \rho)$.
  $(\boldsymbol{x}, \boldsymbol{y}) = (1 - \varepsilon) \cdot (\boldsymbol{x}, \boldsymbol{y}) + \varepsilon \cdot (\boldsymbol{x}', \boldsymbol{y}')$.
Return $(\boldsymbol{x}, \boldsymbol{y})$.

---
**Algorithm 2: FIND_DIRECTION$(\boldsymbol{x}, \boldsymbol{y}, \rho)$**

INPUT: A strategy profile $(\boldsymbol{x}, \boldsymbol{y})$ and parameter $\rho \in (0, 1]$.
OUTPUT: The direction $(\boldsymbol{x}', \boldsymbol{y}')$ that minimizes the $\rho$-directional derivative.

minimize $\gamma$
  s.t. $\gamma \geq (\boldsymbol{e}_i)^\top \boldsymbol{R}\boldsymbol{y}' - (\boldsymbol{x}')^\top \boldsymbol{R}\boldsymbol{e}_j$, for any $i \in BR_r^\rho(\boldsymbol{y})$, for any $j \in BR_c^\rho(\boldsymbol{x})$, and
  with $\boldsymbol{x}', \boldsymbol{y}' \in \Delta^{n-1}$.

Return $(\boldsymbol{x}', \boldsymbol{y}')$.

---

Algorithm 1 takes as input a game and 3 parameters, namely $\delta \in (0, 1]$, which refers to the approximation guarantee that is desired, $\rho \in (0, 1]$ which involves the approximation to the directional derivative, and $\epsilon$, which refers to the size of the step taken in each iteration. We will see shortly that our theoretical analysis requires $\rho$ and $\epsilon$ to be correlated.

**Remark 1.** *The choice of $\rho$ demonstrates the trade off between global optimization (Linear Programming) and the descent-based approach. In the extreme case where $\rho = 1$, then the method will stop after 1 iteration, since it is solving the linear program of the entire zero-sum game (because every pure strategy would belong to $BR_r^\rho(\boldsymbol{y})$). On the other hand, when $\rho$ is small, close to $0$, then the method solves in each iteration rather small linear programs in Algorithm 2 (dependent on the sets $BR_c^\rho(\boldsymbol{x}), BR_r^\rho(\boldsymbol{y})$).*

## 4.2 PROOF OF CORRECTNESS AND RATE OF CONVERGENCE

Our main result is the following theorem.

**Theorem 5.** *For any constants $\delta > 0$, $\rho > 0$, and with $\epsilon = \rho/2$, Algorithm 1 returns a $\delta$-Nash equilibrium in bilinear zero-sum games, after at most $O(\frac{1}{\rho \cdot \delta} \log \frac{1}{\delta})$ iterations.*

In order to prove this Theorem, we will start first with the following auxiliary Lemma.

**Lemma 4.** *If $\varepsilon \leq \frac{\rho}{2}$, then it holds that*

$$\max\left\{0, \max_{i \in BR_r^\rho(\boldsymbol{y})} \boldsymbol{e}_i^\top \boldsymbol{R}\Big((1-\varepsilon)\cdot\boldsymbol{y} + \varepsilon\cdot\boldsymbol{y}'\Big) - \max_{i \in BR_r^\rho(\boldsymbol{y})} \boldsymbol{e}_i^\top \boldsymbol{R}\Big((1-\varepsilon)\cdot\boldsymbol{y} + \varepsilon\cdot\boldsymbol{y}'\Big)\right\} = 0.$$

*Similarly, for the column player, it holds that*

$$\max\left\{0, -\min_{j \in BR_c^\rho(\boldsymbol{x})} \Big((1-\varepsilon)\cdot\boldsymbol{x} + \varepsilon\cdot\boldsymbol{x}'\Big)^\top R\boldsymbol{e}_j + \min_{j \in BR_c^\rho(\boldsymbol{x})} \Big((1-\varepsilon)\cdot\boldsymbol{x} + \varepsilon\cdot\boldsymbol{x}'\Big)^\top R\boldsymbol{e}_j\right\} = 0.$$

Given the previous Lemma, we can now establish that the duality gap decreases geometrically, as long as we have not yet found a $\delta$-approximate equilibrium.

**Lemma 5.** *Let $\epsilon \leq \frac{\rho}{2}$ and suppose that after $t$ iterations we are at a profile $(\boldsymbol{x}^t, \boldsymbol{y}^t)$, which is not a $\delta$-Nash equilibrium. Then,*

$$V(\boldsymbol{x}^{t+1}, \boldsymbol{y}^{t+1}) \leq \left(1 - \frac{\rho\cdot\delta}{4}\right)\cdot V(\boldsymbol{x}^t, \boldsymbol{y}^t),$$

*where $(\boldsymbol{x}^{t+1}, \boldsymbol{y}^{t+1})$ is the strategy profile at $t+1$ iteration.*

*Proof.* To simplify notation, let $\boldsymbol{x}^t = \boldsymbol{x}$, $\boldsymbol{y}^t = \boldsymbol{y}$. Since we move with a step of $\varepsilon$ to the best direction $(\boldsymbol{x}', \boldsymbol{y}')$, by definition we have that

$$(\boldsymbol{x}^{t+1}, \boldsymbol{y}^{t+1}) = ((1-\varepsilon)\cdot\boldsymbol{x} + \varepsilon\cdot\boldsymbol{x}', (1-\varepsilon)\cdot\boldsymbol{y} + \varepsilon\cdot\boldsymbol{y}').$$

We have that

$$V((1-\varepsilon)\cdot\boldsymbol{x} + \varepsilon\cdot\boldsymbol{x}', (1-\varepsilon)\cdot\boldsymbol{y} + \varepsilon\cdot\boldsymbol{y}')$$
$$= \max_i \boldsymbol{e}_i^\top \boldsymbol{R}\Big((1-\varepsilon)\cdot\boldsymbol{y} + \varepsilon\cdot\boldsymbol{y}'\Big) - \min_j \Big((1-\varepsilon)\cdot\boldsymbol{x} + \varepsilon\cdot\boldsymbol{x}'\Big)^\top R\boldsymbol{e}_j. \tag{1}$$

Similar to (2), we have that

$$\max_i \boldsymbol{e}_i^\top \boldsymbol{R}\Big((1-\varepsilon)\cdot\boldsymbol{y} + \varepsilon\cdot\boldsymbol{y}'\Big) = \max_{i \in BR_r^\rho(\boldsymbol{y})} \boldsymbol{e}_i^\top \boldsymbol{R}\Big((1-\varepsilon)\cdot\boldsymbol{y} + \varepsilon\cdot\boldsymbol{y}'\Big)$$
$$+ \max\left\{0, \max_{i \in \overline{BR_r^\rho(\boldsymbol{y})}} \boldsymbol{e}_i^\top \boldsymbol{R}\Big((1-\varepsilon)\cdot\boldsymbol{y} + \varepsilon\cdot\boldsymbol{y}'\Big) - \max_{i \in BR_r^\rho(\boldsymbol{y})} \boldsymbol{e}_i^\top \boldsymbol{R}\Big((1-\varepsilon)\cdot\boldsymbol{y} + \varepsilon\cdot\boldsymbol{y}'\Big)\right\}.$$

Note that since $\varepsilon \leq \frac{\rho}{2}$, Lemma 4 implies

$$\max\left\{0, \max_{i \in \overline{BR_r^\rho(\boldsymbol{y})}} \boldsymbol{e}_i^\top \boldsymbol{R}\Big((1-\varepsilon)\cdot\boldsymbol{y} + \varepsilon\cdot\boldsymbol{y}'\Big) - \max_{i \in BR_r^\rho(\boldsymbol{y})} \boldsymbol{e}_i^\top \boldsymbol{R}\Big((1-\varepsilon)\cdot\boldsymbol{y} + \varepsilon\cdot\boldsymbol{y}'\Big)\right\} = 0.$$

Therefore, it holds that $\max_i \boldsymbol{e}_i^\top \boldsymbol{R}\Big((1-\varepsilon)\cdot\boldsymbol{y} + \varepsilon\cdot\boldsymbol{y}'\Big) = \max_{i \in BR_r^\rho(\boldsymbol{y})} \boldsymbol{e}_i^\top \boldsymbol{R}\Big((1-\varepsilon)\cdot\boldsymbol{y} + \varepsilon\cdot\boldsymbol{y}'\Big)$.

We get a similar analysis for the minimum part in (1), to obtain that $\min_j \Big((1-\varepsilon)\cdot\boldsymbol{x} + \varepsilon\cdot\boldsymbol{x}'\Big)^\top R\boldsymbol{e}_j = \min_{j \in BR_c^\rho(\boldsymbol{x})} \Big((1-\varepsilon)\cdot\boldsymbol{x} + \varepsilon\cdot\boldsymbol{x}'\Big)^\top R\boldsymbol{e}_j$.

Hence, in total we have that

$$V((1-\varepsilon)\cdot\boldsymbol{x} + \varepsilon\cdot\boldsymbol{x}', (1-\varepsilon)\cdot\boldsymbol{y} + \varepsilon\cdot\boldsymbol{y}')$$
$$= \max_{i \in BR_r^\rho(\boldsymbol{y})} \boldsymbol{e}_i^\top \boldsymbol{R}\Big((1-\varepsilon)\cdot\boldsymbol{y} + \varepsilon\cdot\boldsymbol{y}'\Big) - \min_{j \in BR_c^\rho(\boldsymbol{x})} \Big((1-\varepsilon)\cdot\boldsymbol{x} + \varepsilon\cdot\boldsymbol{x}'\Big)^\top R\boldsymbol{e}_j$$
$$\leq (1-\varepsilon)\cdot\max_i \boldsymbol{e}_i^\top \boldsymbol{R}\boldsymbol{y} + \varepsilon\cdot\max_{i \in BR_r^\rho(\boldsymbol{y})} \boldsymbol{e}_i^\top \boldsymbol{R}\boldsymbol{y}' - (1-\varepsilon)\cdot\min_j (\boldsymbol{x})^\top R\boldsymbol{e}_j - \varepsilon\cdot\min_{j \in BR_c^\rho(\boldsymbol{x})} (\boldsymbol{x}')^\top R\boldsymbol{e}_j$$
$$= \max_i \boldsymbol{e}_i^\top \boldsymbol{R}\boldsymbol{y} - \min_j (\boldsymbol{x})^\top R\boldsymbol{e}_j$$
$$+ \varepsilon\cdot\left(\max_{i \in BR_r^\rho(\boldsymbol{y})} \boldsymbol{e}_i^\top \boldsymbol{R}\boldsymbol{y}' - \min_{j \in BR_c^\rho(\boldsymbol{x})} (\boldsymbol{x}')^\top R\boldsymbol{e}_j - \max_i \boldsymbol{e}_i^\top \boldsymbol{R}\boldsymbol{y} + \min_j (\boldsymbol{x})^\top R\boldsymbol{e}_j\right)$$
$$= V(\boldsymbol{x}, \boldsymbol{y}) + \varepsilon\cdot\nabla_{(\boldsymbol{x}',\boldsymbol{y}')} V(\boldsymbol{x}, \boldsymbol{y})$$
$$\leq V(\boldsymbol{x}, \boldsymbol{y}) + \varepsilon\cdot\rho\cdot\nabla_{(\boldsymbol{x}',\boldsymbol{y}')} V(\boldsymbol{x}, \boldsymbol{y}) < V(\boldsymbol{x}, \boldsymbol{y}) - \varepsilon\cdot\delta = (1-c)\cdot V(\boldsymbol{x}, \boldsymbol{y}),$$

with $c = \frac{\varepsilon \cdot \delta}{V(\boldsymbol{x}, \boldsymbol{y})} \geq \frac{\rho \cdot \delta}{4}$, since $V(\boldsymbol{x}, \boldsymbol{y}) \leq 2$ and $\varepsilon = \frac{\rho}{2}$. The second last inequality holds by Lemma 2 and the last since $\rho \cdot \nabla_{(\boldsymbol{x}', \boldsymbol{y}')} V(\boldsymbol{x}, \boldsymbol{y}) < -\delta$, since we are not in a $\delta$-NE. $\qquad\square$

Finally, we are ready to complete the proof of our main theorem.

**Proof of Theorem 5.** We want to find the number of iterations to reach a $\delta$-Nash equilibrium. Suppose that we manage to achieve this at iteration $t$, with profile $(\boldsymbol{x}^t, \boldsymbol{y}^t)$. By Lemma 5, we have that

$$V(\boldsymbol{x}^t, \boldsymbol{y}^t) \leq (1-c) \cdot V(\boldsymbol{x}^{t-1}, \boldsymbol{y}^{t-1}) \leq \cdots \leq (1-c)^t \cdot V(\boldsymbol{x}^0, \boldsymbol{y}^0)$$

with $c = \frac{\rho \cdot \delta}{4}$ be constant. In order to ensure that this is bounded by $\delta$, it suffices to have that $2 \cdot (1-c)^t \leq \delta$, since $V(\boldsymbol{x}^0, \boldsymbol{y}^0) \leq 2$. This means that we need $(1-c)^t \leq \frac{\delta}{2} \Rightarrow t \cdot \log(1-c) \leq \log \frac{\delta}{2}$. This implies that $t \cdot \log \frac{1}{(1-c)} \geq \log \frac{2}{\delta}$. Hence, we can see that the following lower bound should hold for $t$.

$$t \geq \frac{\log \frac{2}{\delta}}{\log \frac{1}{(1-c)}} \geq \frac{\log \frac{2}{\delta}}{\log \frac{1}{(1-\frac{\rho \cdot \delta}{4})}} = \frac{\log \frac{2}{\delta}}{\log \frac{4}{4-\rho \cdot \delta}} \geq \frac{\log \frac{2}{\delta}}{\frac{4}{4-\rho \cdot \delta} - 1} = \frac{\log \frac{2}{\delta}}{\frac{\rho \cdot \delta}{4-\rho \cdot \delta}} = \frac{4 - \rho \cdot \delta}{\rho \cdot \delta} \cdot \log \frac{2}{\delta} \geq \frac{4}{\rho \cdot \delta} \cdot \log \frac{2}{\delta}$$

The last but one inequality holds since $\frac{1}{\log x} \geq \frac{1}{x-1}$, for $x \geq 1$. $\qquad\square$

### 4.3 Speeding up the algorithm via a decaying schedule

In this section, we will improve the convergence guarantee by using the developed algorithm of the previous section as a black-box. The idea is to gradually decaying $\delta$ and use it to bound $c$, instead of the more coarse approximation of $V(\boldsymbol{x}, \boldsymbol{y}) \leq 2$.

---

**Decaying Delta Speedup: Algorithm 3.**

INPUT: A zero-sum game $(\boldsymbol{R}, -\boldsymbol{R})$, and an approximation $\delta > 0$ be a fixed constant and $\rho > 0$ a fixed constant.

OUTPUT: A $\delta$-NE strategy profile.

> Pick an arbitrary strategy profile $(\boldsymbol{x}, \boldsymbol{y})$.
> Set $i = 0$, $\delta_0 = \frac{V(\boldsymbol{x}, \boldsymbol{y})}{2}$, $\varepsilon = \frac{\rho}{2}$.
> Do
> > Update $(\boldsymbol{x}, \boldsymbol{y})$ via Algorithm 1 $\left( (\boldsymbol{R}, -\boldsymbol{R}), \delta_i, \rho, \varepsilon \right)$.
> > $i = i + 1$, $\delta_i = \delta_i / 2$.
> > While $\delta_i > \delta$
>
> Return $(\boldsymbol{x}, \boldsymbol{y})$.

---

**Theorem 6.** *Algorithm 3 reaches a $\delta$-Nash equilibrium after at most $O\left( \frac{1}{\rho} \cdot \log\left( \frac{1}{\delta} \right) \right)$ iterations, for any constant $\delta$.*

*Proof.* Consider an iteration of the outer loop with $\delta_i$. In this case, for any $i > 0$, we have that $c_i \geq \frac{\rho \cdot \delta_i}{2 \cdot V(\boldsymbol{x}, \boldsymbol{y})} \geq \frac{\rho \cdot \delta_i}{2 \cdot \delta_{i-1}} = \frac{\rho \cdot \delta_i}{4 \cdot \delta_i} = \frac{\rho}{4}$, since $V(x, y) \leq \delta_{i-1} = 2 \cdot \delta_i$ be the duality gap at the beginning of the epoch. Furthermore, we can see that since $V(\boldsymbol{x}, \boldsymbol{y}) \leq 2$ and $\delta_0$ be a constant we have $c_0 \geq \frac{\rho \cdot \delta_0}{4}$ be a constant for $\rho$ and $\delta_0$ constants, which implies that $t_0$ is a constant. Let now $t_i$, for any $i > 0$, be the number of iterations of Algorithm 1 in the current epoch to achieve a $\delta_i$-NE,

then, similar to the proof of Theorem 5, we have that

$$(1 - c_i)^{t_i} \cdot \delta_{i-1} \leq \delta_i \implies (1 - c_i)^{t_i} \cdot 2 \cdot \delta_i \leq \delta_i \implies (1 - c_i)^{t_i} \leq \frac{1}{2}$$

$$\implies t_i \cdot \log\left(\frac{4 - \rho}{4}\right) \leq \log 1/2 \implies t_i \geq \frac{1}{\log\left(\frac{4}{4-\rho}\right)} \geq \frac{4 - \rho}{\rho}.$$

which implies that $t_i = O(\frac{1}{\rho})$, since $c_i$ is a constant. Note now that the maximum total number $k$ of epochs to achieve a $\delta$-NE is $\frac{\delta_0}{2^k} = \delta \to k = O\left(\log\left(\frac{\delta_0}{\delta}\right)\right)$. Thus, the total number of iterations are

$$t = \sum_{i=0:k} t_i = \sum_{i=0:k} O\left(\frac{1}{\rho}\right) = (k+1) \cdot O\left(\frac{1}{\rho}\right) = O\left(\frac{1}{\rho} \cdot \log\left(\frac{1}{\delta}\right)\right).$$

$\square$

## 5 EXPERIMENTAL EVALUATION

In this section we present some experimental evidence regarding the actual performance of the method in practice.

**Experimental Setup**  For values of $n$ ranging from 100 to 1000, we generated random $n \times n$ zero-sum games, where each entry was selected uniformly at random from $[0, 1]$. We derived at least 10 random games for each value of $n$ that we considered. We also experimented with several values for the parameters $\delta$, $\rho$ and $\epsilon$. Further tuning of these parameters, which we leave as future work, may provide even better running times.

**Number of iterations.**  In Figure 1, we demonstrate the behavior we observed, where for each value of $n$, the average number of iterations over the games produced is depicted. What we see is that even though there is some increase on the required number of iterations to reach an approximate equilibrium, as $n$ becomes larger, this increase is quite moderate (the actual time needed per iteration does increase though, as one needs to deal with larger LPs).

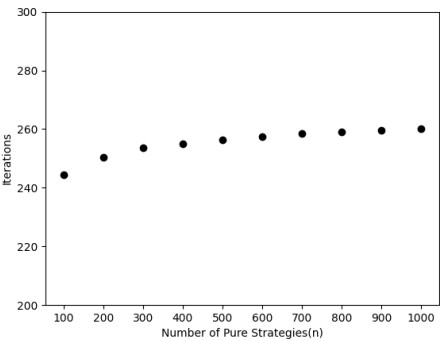

Figure 1: Number of iterations as $n$ grows, with $\rho = 0.05$, $\delta = 0.001$

**Size of LPs in Algorithm 2.**  A crucial aspect of our method is that in each iteration, Algorithm 2 is required to solve a LP. In Figures 2 and 3, we depict the number of strategies that participate in the best response sets, more precisely in the sets $BR_c^\rho(\boldsymbol{x}), BR_r^\rho(\boldsymbol{y})$, which in turn affect the number of constraints in these LPs. The figures we show the size of the best response sets as a percentage of $n$, averaged over the two players. As we see, the approximate best response sets are sparse, which is a positive aspect of our method. In Figure 2 we see how this percentage varies as we decrease the required approximation $\delta$, for a fixed dimension $n = 1000$, whereas in Figure 3, we see how the sparsity varies as $n$ grows.

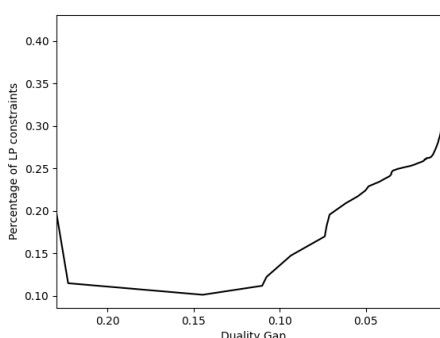
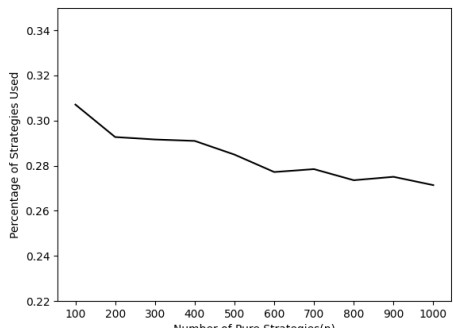

Figure 2: Percentage as $\delta$ goes to 0, with $n = 1000$, $\rho = 0.05$

Figure 3: Percentage as $n$ grows with $\delta = 0.001$, $\rho = 0.05$

| $n$ | Ours | IPM |
|---|---|---|
| 500 | 0.03 | 0.05 |
| 1000 | 0.11 | 0.28 |
| 1500 | 0.3 | 0.73 |
| 2000 | 0.6 | 1.5 |

Table 1: Running time comparisons in seconds against the SciPy interior point method.

**Running time comparisons with standard solvers.** Although our method does not outperform standard solvers in general, we have seen classes of games where our running time is comparable or even better. One such class is the family of block games where we have non-zero entries in the upper left and the right bottom part of the matrix. Indicatively we provide a summary on the table above.

## 6 CONCLUSIONS

We have analyzed a descent-based method for the duality gap in zero-sum games. In this work our goal has been to demonstrate our method, based on the descent on the duality gap, as a proof of concept. We expect that our method can be further optimized in practice and find this a promising direction for future work. In particular, one idea to explore is whether we can reuse the LP solutions we get in Algorithm 2 from one iteration to the next (since we only change the current solution slightly by a step of size $\epsilon$). Exploring such *warm start* strategies (see e.g. Yildirim & Wright (2002)) could provide significant speedups. An additional idea for acceleration is to utilize results regarding low support approximate equilibria, hence reducing further the size (the number of LP constraints, and thus number of participating rows and columns in each iteration of Algorithm 2). We can for example reduce $\rho$ according to some schedule so that the best response sets BR become smaller in size.

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

## A    MISSING PROOFS FROM SECTION 4

### A.1    PROOF OF THEOREM 4

Using the definition of the duality gap, the directional derivative is equal to

$$\nabla_{(\boldsymbol{x}', \boldsymbol{y}')} V(\boldsymbol{x}, \boldsymbol{y})$$

$$= \lim_{\varepsilon \searrow 0} \frac{\max_i \boldsymbol{e}_i^\top \boldsymbol{R} \Big( (1 - \varepsilon) \cdot \boldsymbol{y} + \varepsilon \cdot \boldsymbol{y}' \Big) - \min_j \Big( (1 - \varepsilon) \cdot \boldsymbol{x} + \varepsilon \cdot \boldsymbol{x}' \Big)^\top \boldsymbol{R} \boldsymbol{e}_j - V(\boldsymbol{x}, \boldsymbol{y})}{\varepsilon}.$$

However, we can write up the term $\max_i \boldsymbol{e}_i^\top \boldsymbol{R} \Big( (1 - \varepsilon) \cdot \boldsymbol{y} + \varepsilon \cdot \boldsymbol{y}' \Big)$, similarly to Deligkas et al. (2017); Tsaknakis & Spirakis (2008), as

$$\max_i \boldsymbol{e}_i^\top \boldsymbol{R} \Big( (1 - \varepsilon) \cdot \boldsymbol{y} + \varepsilon \cdot \boldsymbol{y}' \Big) = \max_{i \in BR_r(\boldsymbol{y})} \boldsymbol{e}_i^\top \boldsymbol{R} \Big( (1 - \varepsilon) \cdot \boldsymbol{y} + \varepsilon \cdot \boldsymbol{y}' \Big)$$

$$+ \max \Big\{ 0, \max_{i \in \overline{BR_r(\boldsymbol{y})}} \boldsymbol{e}_i^\top \boldsymbol{R} \Big( (1 - \varepsilon) \cdot \boldsymbol{y} + \varepsilon \cdot \boldsymbol{y}' \Big) - \max_{i \in BR_r(\boldsymbol{y})} \boldsymbol{e}_i^\top \boldsymbol{R} \Big( (1 - \varepsilon) \cdot \boldsymbol{y} + \varepsilon \cdot \boldsymbol{y}' \Big) \Big\}, \tag{2}$$

where $\overline{BR_r(\boldsymbol{y})}$ is the complement set of $BR_r(\boldsymbol{y})$. Similarly,

$$\min_j \Big( (1 - \varepsilon) \cdot \boldsymbol{x} + \varepsilon \cdot \boldsymbol{x}' \Big)^\top R \boldsymbol{e}_j = \min_{j \in BR_c(\boldsymbol{x})} \Big( (1 - \varepsilon) \cdot \boldsymbol{x} + \varepsilon \cdot \boldsymbol{x}' \Big)^\top R \boldsymbol{e}_j$$

$$- \max \Big\{ 0, - \min_{j \in \overline{BR_c(\boldsymbol{x})}} \Big( (1 - \varepsilon) \cdot \boldsymbol{x} + \varepsilon \cdot \boldsymbol{x}' \Big)^\top R \boldsymbol{e}_j + \min_{j \in BR_c(\boldsymbol{x})} \Big( (1 - \varepsilon) \cdot \boldsymbol{x} + \varepsilon \cdot \boldsymbol{x}' \Big)^\top R \boldsymbol{e}_j \Big\}. \tag{3}$$

Arguing in a similar fashion as in Deligkas et al. (2017), there exists $\epsilon^* > 0$ such that for $\epsilon \leq \epsilon^*$, the term

$$\max \Big\{ 0, \max_{i \in \overline{BR_r(\boldsymbol{y})}} \boldsymbol{e}_i^\top \boldsymbol{R} \Big( (1 - \varepsilon) \cdot \boldsymbol{y} + \varepsilon \cdot \boldsymbol{y}' \Big) - \max_{i \in BR_r(\boldsymbol{y})} \boldsymbol{e}_i^\top R \Big( (1 - \varepsilon) \cdot \boldsymbol{y} + \varepsilon \cdot \boldsymbol{y}' \Big) \Big\}$$

is zero, and hence it can be ignored when we take the limit of $\epsilon \to 0$. In the same manner, the corresponding term for the column player also becomes 0. Note also that for any $i, j \in BR_r(\boldsymbol{y})$, we have that $\boldsymbol{e}_i^\top \boldsymbol{R} \boldsymbol{y} = \boldsymbol{e}_j^\top \boldsymbol{R} \boldsymbol{y}$, and hence the term $\max_{i \in BR_r(\boldsymbol{y})} \boldsymbol{e}_i^\top \boldsymbol{R} \boldsymbol{y}$ is independent of the row we choose, thus $\max_{i \in BR_r(\boldsymbol{y})} \boldsymbol{e}_i^\top \boldsymbol{R} \Big( (1 - \varepsilon) \cdot \boldsymbol{y} + \varepsilon \cdot \boldsymbol{y}' \Big) = (1 - \varepsilon) \cdot \max_{i \in BR_r(\boldsymbol{y})} \boldsymbol{e}_i^\top \boldsymbol{R} \boldsymbol{y} + \varepsilon \cdot \max_{i \in BR_r(\boldsymbol{y})} \boldsymbol{e}_i^\top \boldsymbol{R} \boldsymbol{y}'$, similar for the $\min$ part. By using this below, we conclude that the directional derivative equals to

$$\lim_{\varepsilon \searrow 0} \frac{\max_{i \in BR_r(\boldsymbol{y})} \boldsymbol{e}_i^\top \boldsymbol{R} \Big( (1 - \varepsilon) \cdot \boldsymbol{y} + \varepsilon \cdot \boldsymbol{y}' \Big) - \min_{j \in BR_c(\boldsymbol{x})} \Big( (1 - \varepsilon) \cdot \boldsymbol{x} + \varepsilon \cdot \boldsymbol{x}' \Big)^\top \boldsymbol{R} \boldsymbol{e}_j - V(\boldsymbol{x}, \boldsymbol{y})}{\varepsilon}$$

$$= \lim_{\varepsilon \searrow 0} \frac{(1 - \varepsilon) \cdot \max_{i \in BR_r(\boldsymbol{y})} \boldsymbol{e}_i^\top \boldsymbol{R} \boldsymbol{y} + \varepsilon \cdot \max_{i \in BR_r(\boldsymbol{y})} \boldsymbol{e}_i^\top \boldsymbol{R} \boldsymbol{y}'}{\varepsilon}$$

$$- \frac{(1 - \varepsilon) \cdot \min_{j \in BR_c(\boldsymbol{x})} x^\top \boldsymbol{R} \boldsymbol{e}_j + \varepsilon \cdot \min_{j \in BR_c(\boldsymbol{x})} (\boldsymbol{x}')^\top \boldsymbol{R} \boldsymbol{e}_j + V(\boldsymbol{x}, \boldsymbol{y})}{\varepsilon}$$

$$= \lim_{\varepsilon \searrow 0} \frac{(1 - \varepsilon) \cdot V(\boldsymbol{x}, \boldsymbol{y}) + \varepsilon \cdot \Big( \max_{i \in BR_r(\boldsymbol{y})} \boldsymbol{e}_i^\top \boldsymbol{R} \boldsymbol{y}' - \min_{j \in BR_c(\boldsymbol{x})} (\boldsymbol{x}')^\top \boldsymbol{R} \boldsymbol{e}_j \Big) - V(\boldsymbol{x}, \boldsymbol{y})}{\varepsilon}$$

$$= \max_{i \in BR_r(\boldsymbol{y})} \boldsymbol{e}_i^\top \boldsymbol{R} \boldsymbol{y}' - \min_{j \in BR_c(\boldsymbol{x})} (\boldsymbol{x}')^\top \boldsymbol{R} \boldsymbol{e}_j - V(\boldsymbol{x}, \boldsymbol{y}).$$

as claimed.

## A.2 Proof of Lemma 4

Firstly, we have that

$$\max_{i \in BR_r^\rho(\boldsymbol{y})} \boldsymbol{e}_i^\top \boldsymbol{R}\Big((1-\varepsilon)\cdot\boldsymbol{y}+\varepsilon\cdot\boldsymbol{y}'\Big) \geq \max_{i \in BR_r^\rho(\boldsymbol{y})} \boldsymbol{e}_i^\top \boldsymbol{R}\Big((1-\varepsilon)\cdot\boldsymbol{y}\Big)$$
$$= (1-\varepsilon)\cdot \max_{i \in BR_r^\rho(\boldsymbol{y})} \boldsymbol{e}_i^\top \boldsymbol{R}\boldsymbol{y} = \max_{i \in BR_r^\rho(\boldsymbol{y})} \boldsymbol{e}_i^\top \boldsymbol{R}\boldsymbol{y} - \varepsilon\cdot \max_{i \in BR_r^\rho(\boldsymbol{y})} \boldsymbol{e}_i^\top \boldsymbol{R}\boldsymbol{y}.$$

By the definition of the $\max$ function, we have

$$\max_{i \in \overline{BR_r^\rho(\boldsymbol{y})}} \boldsymbol{e}_i^\top \boldsymbol{R}\Big((1-\varepsilon)\cdot\boldsymbol{y}+\varepsilon\cdot\boldsymbol{y}'\Big) \leq (1-\varepsilon)\cdot \max_{i \in \overline{BR_r^\rho(\boldsymbol{y})}} \boldsymbol{e}_i^\top \boldsymbol{R}\boldsymbol{y} + \varepsilon\cdot \max_{i \in \overline{BR_r^\rho(\boldsymbol{y})}} \boldsymbol{e}_i^\top \boldsymbol{R}\boldsymbol{y}'.$$

These two bounds give

$$\max_{i \in \overline{BR_r^\rho(\boldsymbol{y})}} \boldsymbol{e}_i^\top \boldsymbol{R}\Big((1-\varepsilon)\cdot\boldsymbol{y}+\varepsilon\cdot\boldsymbol{y}'\Big) - \max_{i \in BR_r^\rho(\boldsymbol{y})} \boldsymbol{e}_i^\top \boldsymbol{R}\Big((1-\varepsilon)\cdot\boldsymbol{y}+\varepsilon\cdot\boldsymbol{y}'\Big)$$
$$\leq \max_{i \in \overline{BR_r^\rho(\boldsymbol{y})}} \boldsymbol{e}_i^\top \boldsymbol{R}\boldsymbol{y} + \varepsilon\cdot\Big(\max_{i \in \overline{BR_r^\rho(\boldsymbol{y})}} \boldsymbol{e}_i^\top \boldsymbol{R}\boldsymbol{y}' - \max_{i \in \overline{BR_r^\rho(\boldsymbol{y})}} \boldsymbol{e}_i^\top \boldsymbol{R}\boldsymbol{y}\Big)$$
$$- \max_{i \in BR_r^\rho(\boldsymbol{y})} \boldsymbol{e}_i^\top \boldsymbol{R}\boldsymbol{y} + \varepsilon\cdot \max_{i \in BR_r^\rho(\boldsymbol{y})} \boldsymbol{e}_i^\top \boldsymbol{R}\boldsymbol{y}$$
$$= \max_{i \in \overline{BR_r^\rho(\boldsymbol{y})}} \boldsymbol{e}_i^\top \boldsymbol{R}\boldsymbol{y} - \max_{i \in BR_r^\rho(\boldsymbol{y})} \boldsymbol{e}_i^\top \boldsymbol{R}\boldsymbol{y} + \varepsilon\cdot\Big(\max_{i \in \overline{BR_r^\rho(\boldsymbol{y})}} \boldsymbol{e}_i^\top \boldsymbol{R}\boldsymbol{y}' - \max_{i \in \overline{BR_r^\rho(\boldsymbol{y})}} \boldsymbol{e}_i^\top \boldsymbol{R}\boldsymbol{y} + \max_{i \in BR_r^\rho(\boldsymbol{y})} \boldsymbol{e}_i^\top \boldsymbol{R}\boldsymbol{y}\Big).$$

By the definition of $\rho$-best-response, we have that

$$\max_{i \in \overline{BR_r^\rho(\boldsymbol{y})}} \boldsymbol{e}_i^\top \boldsymbol{R}\boldsymbol{y} - \max_{i \in BR_r^\rho(\boldsymbol{y})} \boldsymbol{e}_i^\top \boldsymbol{R}\boldsymbol{y} < -\rho.$$

Furthermore, we have that $\max_{i \in \overline{BR_r^\rho(\boldsymbol{y})}} \boldsymbol{e}_i^\top \boldsymbol{R}\boldsymbol{y}' - \max_{i \in \overline{BR_r^\rho(\boldsymbol{y})}} \boldsymbol{e}_i^\top \boldsymbol{R}\boldsymbol{y} + \max_{i \in BR_r^\rho(\boldsymbol{y})} \boldsymbol{e}_i^\top \boldsymbol{R}\boldsymbol{y} \leq 2$, since $R_{ij} \leq 1$. Thus, we have that

$$\max_{i \in \overline{BR_r^\rho(\boldsymbol{y})}} \boldsymbol{e}_i^\top \boldsymbol{R}\boldsymbol{y} - \max_{i \in BR_r^\rho(\boldsymbol{y})} \boldsymbol{e}_i^\top \boldsymbol{R}\boldsymbol{y} + \varepsilon\cdot\Big(\max_{i \in \overline{BR_r^\rho(\boldsymbol{y})}} \boldsymbol{e}_i^\top \boldsymbol{R}\boldsymbol{y}' - \max_{i \in \overline{BR_r^\rho(\boldsymbol{y})}} \boldsymbol{e}_i^\top \boldsymbol{R}\boldsymbol{y} + \max_{i \in BR_r^\rho(\boldsymbol{y})} \boldsymbol{e}_i^\top \boldsymbol{R}\boldsymbol{y}\Big)$$
$$< -\rho + 2\cdot\varepsilon.$$

So, we want to find a value of $\varepsilon$ such that $-\rho + 2\cdot\varepsilon \leq 0$, which holds for $\varepsilon \leq \frac{\rho}{2}$. In a very similar fashion, for the second part of the Lemma, we have

$$\min_{j \in \overline{BR_c^\rho(\boldsymbol{x})}} \Big((1-\varepsilon)\cdot\boldsymbol{x}+\varepsilon\cdot\boldsymbol{x}'\Big)^\top R\boldsymbol{e}_j \geq (1-\varepsilon)\cdot \min_{j \in \overline{BR_c^\rho(\boldsymbol{x})}} \boldsymbol{x}^\top R\boldsymbol{e}_j + \varepsilon\cdot \min_{j \in \overline{BR_c^\rho(\boldsymbol{x})}} (\boldsymbol{x}')^\top R\boldsymbol{e}_j.$$

Furthermore,

$$\min_{j \in BR_c^\rho(\boldsymbol{x})} \Big((1-\varepsilon)\cdot\boldsymbol{x}+\varepsilon\cdot\boldsymbol{x}'\Big)^\top R\boldsymbol{e}_j = -\max_{j \in BR_c^\rho(\boldsymbol{x})} \Big((1-\varepsilon)\cdot\boldsymbol{x}+\varepsilon\cdot\boldsymbol{x}'\Big)^\top (-R)\boldsymbol{e}_j$$
$$\leq -(1-\varepsilon)\cdot \max_{j \in BR_c^\rho(\boldsymbol{x})} \boldsymbol{x}^\top (-R)\boldsymbol{e}_j = (1-\varepsilon)\cdot \min_{j \in BR_c^\rho(\boldsymbol{x})} \boldsymbol{x}^\top R\boldsymbol{e}_j.$$

The above inequality holds since $\max_{j \in BR_c^\rho(\boldsymbol{x})} \Big((1-\varepsilon)\cdot\boldsymbol{x}+\varepsilon\cdot\boldsymbol{x}'\Big)^\top (-R)\boldsymbol{e}_j \geq \max_{j \in BR_c^\rho(\boldsymbol{x})} \Big((1-\varepsilon)\cdot\boldsymbol{x}\Big)^\top (-R)\boldsymbol{e}_j = (1-\varepsilon)\cdot\max_{j \in BR_c^\rho(\boldsymbol{x})} \boldsymbol{x}^\top (-R)\boldsymbol{e}_j$. Thus, these two bounds give

$$-\min_{j \in \overline{BR_c^\rho(\boldsymbol{x})}} \Big((1-\varepsilon)\cdot\boldsymbol{x}+\varepsilon\cdot\boldsymbol{x}'\Big)^\top R\boldsymbol{e}_j + \min_{j \in BR_c^\rho(\boldsymbol{x})} \Big((1-\varepsilon)\cdot\boldsymbol{x}+\varepsilon\cdot\boldsymbol{x}'\Big)^\top R\boldsymbol{e}_j$$
$$\leq -(1-\varepsilon)\cdot \min_{j \in \overline{BR_c^\rho(\boldsymbol{x})}} \boldsymbol{x}^\top R\boldsymbol{e}_j - \varepsilon\cdot \min_{j \in \overline{BR_c^\rho(\boldsymbol{x})}} (\boldsymbol{x}')^\top R\boldsymbol{e}_j + (1-\varepsilon)\cdot \min_{j \in BR_c^\rho(\boldsymbol{x})} \boldsymbol{x}^\top R\boldsymbol{e}_j$$
$$= -\min_{j \in \overline{BR_c^\rho(\boldsymbol{x})}} \boldsymbol{x}^\top R\boldsymbol{e}_j + \min_{j \in BR_c^\rho(\boldsymbol{x})} \boldsymbol{x}^\top R\boldsymbol{e}_j$$
$$+ \varepsilon\cdot\Big(\min_{j \in \overline{BR_c^\rho(\boldsymbol{x})}} \boldsymbol{x}^\top R\boldsymbol{e}_j - \min_{j \in \overline{BR_c^\rho(\boldsymbol{x})}} (\boldsymbol{x}')^\top R\boldsymbol{e}_j - \min_{j \in BR_c^\rho(\boldsymbol{x})} \boldsymbol{x}^\top R\boldsymbol{e}_j\Big).$$

But, by the definition of $\rho$-best-response, we have that

$$- \min_{j \in \overline{BR_c^\rho(\boldsymbol{x})}} \boldsymbol{x}^\top R \boldsymbol{e}_j + \min_{j \in BR_c^\rho(\boldsymbol{x})} \boldsymbol{x}^\top R \boldsymbol{e}_j < -\rho,$$

and

$$\min_{j \in BR_c^\rho(\boldsymbol{x})} \boldsymbol{x}^\top R \boldsymbol{e}_j - \min_{j \in \overline{BR_c^\rho(\boldsymbol{x})}} (\boldsymbol{x}')^\top R \boldsymbol{e}_j - \min_{j \in BR_c^\rho(\boldsymbol{x})} \boldsymbol{x}^\top R \boldsymbol{e}_j \leq 1,$$

since $R_{ij} \leq 1$. Thus, in total we have that

$$- \min_{j \in \overline{BR_c^\rho(\boldsymbol{x})}} \boldsymbol{x}^\top R \boldsymbol{e}_j + \min_{j \in BR_c^\rho(\boldsymbol{x})} \boldsymbol{x}^\top R \boldsymbol{e}_j$$

$$+ \varepsilon \cdot \left( \min_{j \in BR_c^\rho(\boldsymbol{x})} \boldsymbol{x}^\top R \boldsymbol{e}_j - \min_{j \in \overline{BR_c^\rho(\boldsymbol{x})}} (\boldsymbol{x}')^\top R \boldsymbol{e}_j - \min_{j \in BR_c^\rho(\boldsymbol{x})} \boldsymbol{x}^\top R \boldsymbol{e}_j \right) < -\rho + \varepsilon.$$

Thus, we only need to ensure that $-\rho + \varepsilon \leq 0$, which is true when $\varepsilon \leq \frac{\rho}{2}$.