# OpenReview forum: "Finding Equilibria in Bilinear Zero-sum Games via a Convexity-based Approach"
_ICLR.cc/2025/Conference — Submitted to ICLR 2025_

### Official Review · Reviewer_8C9U · 2024-11-01

**Soundness:** 3
**Presentation:** 3
**Contribution:** 2
**Rating:** 3
**Confidence:** 4

**Summary:**

This paper designs an optimization algorithm for computing an $\delta$-approximate Nash equilibrium in two-player zero-sum games. Based on the observation that the duality gap function is convex, they develop a steepest gradient descent type algorithm that minimizes the duality gap. The algorithm needs to solve a linear program (LP) in each iteration to find the descent direction, where the LP is smaller than the generic LP that directly solves the Nash equilibrium. They give a convergence rate of $O(\frac{1}{\rho} \log \frac{1}{\delta})$ for their algorithm, where $\rho \in (0,1]$ controls the size of the LP in each iteration (when $\rho = 1$, the LP becomes the generic LP for NE). They also conduct numerical experiments on random matrices and compared their algorithms with standard LP solvers and found in certain cases; their algorithm is faster.

**Strengths:**

1 The idea of performing gradient descent directly on the duality gap is interesting.
2. The presentation of the paper is clear.

**Weaknesses:**

1. The algorithm lacks a precise computational complexity analysis. Although a $O(\frac{1}{\rho} \log \frac{1}{\delta})$ iteration-complexity is given, this bound is not very informative: one may want to choose $\rho = 1$ to minimize the iteration number needed but it then becomes solve one LP for the NE, equivalent to the LP approach. How to choose $\rho$ is unclear since the per-iteration complexity depends on the $\rho$, which affects the size of LP. It is crucial to provide a precise time-complexity analysis of the algorithm, which helps to understand why this iterative approach by solving a series of smaller LPs might be better than solving a large LP once.
2. This paper focuses on the LP approach for solving NE in zero-sum games. Yet, recently, gradient-based first-order methods have become more popular for solving large-scale LPs and zero-sum games than interior-point methods. These algorithms include Extragradient, Regret Matching+ [1], and Primal-Dual Hybrid Gradient Methods [2]. These algorithms also have instance-dependent linear convergence and only require performing gradient steps in each iteration rather than solving an LP. It would be helpful to add experiments on these methods and compare their performances with the proposed algorithm on large-scale instances.

Minor Comments
1. Page 3, Line 111: "The currently best rate is $O(\sqrt{1/T})$ in terms of the duality gap..." [3] has proposed an algorithm with an accelerated $O(1/T)$ convergence rate.

[1] Tammelin, O., Burch, N., Johanson, M., & Bowling, M. (2015, June). Solving heads-up limit texas hold'em. In Twenty-fourth international joint conference on artificial intelligence.

[2] Lu, Haihao, and Jinwen Yang. "On the Infimal Sub-differential Size of Primal-Dual Hybrid Gradient Method and Beyond." arXiv preprint arXiv:2206.12061 (2022).

[3] Cai, Yang, and Weiqiang Zheng. "Doubly optimal no-regret learning in monotone games." International Conference on Machine Learning. 2023.

**Questions:**

See weakness for details.
1. Could you provide a time-complexity analysis of the proposed algorithm?
2. Could you add numerical experiments and compare other gradient-based algorithms?

---

> ### Author Response · Authors · 2024-11-21
> **Response to Reviewer 8C9U**
>
> We thank the Reviewer 8C9U for the review and the questions.
>
> > 1. Could you provide a time-complexity analysis of the proposed algorithm?
>
> Unfortunately the worst case time-complexity analysis of the algorithm is given by the crude upper bound of $\log(\frac{1}{\delta})$ times the complexity of a single full LP (since in each iteration at worst case, we may need to solve the entire LP). This however is not so meaningful for us as in each iteration we solve much smaller LPs on average, but without any theoretical guarantee on their sizes. To see that the worst case can occur, for an arbitrary $\rho$, consider matrices with entries in $(1-\rho, 1]$ (then all strategies are $\rho$-best responses). Therefore, what would be more meaningful is an average case or smoothed analysis.
>
> > 2. Could you add numerical experiments and compare other gradient-based algorithms?
>
> We will try to add more comparisons with gradient-based methods. In fact we have already made some comparisons with the Optimistic MWU, and we can add the results of these experiments within the next few days. See also our response to Reviewer yN9W.
>
> Finally, we can address the minor comment in an updated version.

---

### Official Review · Reviewer_oqZJ · 2024-11-04

**Soundness:** 3
**Presentation:** 3
**Contribution:** 2
**Rating:** 3
**Confidence:** 4

**Summary:**

The paper introduces a new algorithm for finding equilibria in two-player zero-sum games by applying steepest descent to the duality gap/exploitability.  The authors show it achieves linear convergence in the exploitability/duality gap. Simulations demonstrate that its performance is comparable to the performance of LP solvers on at least some games.

**Strengths:**

The algorithm presented by the paper is interesting, and it is notable that linear convergence can be achieved on the exploitability. It is also interesting that the method appears to generate sparse support in practice.

**Weaknesses:**

While the contributions of the paper are interesting, it is not clear to me that the contributions meet the threshold for publication.

The motivation for the research itself is not very clear to me. It is interesting that the approximation algorithms for general-sum games that do descent on the max regret (and a sort of correction) can be specialized to the duality gap in the two-player zero-sum game case. As the author notes, those works focus on polytime approximation algorithms for general-sum games and don't focus on the descent case, so the work on the steepest descent on the duality gap is novel. And, of course, the setting in which it is realistic to use LPs to compute equilibria is when the matrices are relatively small.

However, for large-scale games, it is not clear how well they would work. If there is hope for it to scale to large-scale games, why restrict the matrix size to 1000, and why not compare to non LP based algorithms (e.g., regret minimization based algorithms)?  To be nitpicky, it seems misleading to mention that it scales to "thousands" of strategies, when you stop at 1000 (even though technically the statement is accurate). On the other hand, if the primary contribution is theoretical and the experimental work is just a proof of concept, the theoretical contribution, while interesting, doesn't seem to be an ICLR publication.

A more thorough review of the literature might be useful for the paper. Additionally, the experimental section could be more thorough Some suggestions and questions are included in the following section.

**Questions:**

1. It seems appropriate to cite and discuss the linear convergence of EG/OG for bilinear saddle-point problems over polyhedral domains based on error bounds (e.g., *On linear convergence of iterative methods for the variational inequality problem* Tseng 1995, *Linear Last-iterate Convergence in Constrained Saddle-point Optimization* Wei et al. ). While you mention Gilpin et al.'s algorithm, the same has been known for VIP (again with a dependence on a condition number associated with the system), and so while it is true that the Cai et al. paper has the SOTA rate for condition-number-free rates for last iterate, the discussion in the optimization section is incomplete.

3. It seems that it would be good to mention explicitly the existence of a direction that minimizes the directional derivative; of course this follows from the fact that the steepest descent computation can be formulated as an LP over a compact polyhedral set. While it is mentioned in line 204 that the direction can be identified by solving an LP, it seems worth explicitly mentioning this after Theorem 4, before in Lemmas 1 and 3 you make references to a direction that minimizes the ($\rho$)-directional derivative.

4. Can you compare to work done using descent methods with the Nikaido-Isoda (NI) function? It seems to be that it might be relevant to mention in related work.

5. There should be more information on exactly what the family of block games looks like and how they are generated. It would be good to show running time results for the classes of games that the method does not do well on as well (seems odd to handpick the class for the timing results).

---

> ### Author Response · Authors · 2024-11-21
> **Response to Reviewer oqZj**
>
> We thank the Reviewer oqZJ for the review and the questions.
>
>  > The motivation for the research itself is not very clear to me. It is interesting that the approximation algorithms for general-sum games that do descent on the max regret (and a sort of correction) can be specialized to the duality gap in the two-player zero-sum game case. As the author notes, those works focus on polytime approximation algorithms for general-sum games and don't focus on the descent case, so the work on the steepest descent on the duality gap is novel. And, of course, the setting in which it is realistic to use LPs to compute equilibria is when the matrices are relatively small.
>
> Our starting point of work was indeed the theoretical side of the paper and the realization that the duality gap is a convex function for 0-sum games (see also our initial comments to Reviewer yN9W). Later on we considered that the experimental results may be of interest as well.
>
>  > However, for large-scale games, it is not clear how well they would work. If there is hope for it to scale to large-scale games, why restrict the matrix size to 1000, and why not compare to non LP based algorithms (e.g., regret minimization based algorithms)? To be nitpicky, it seems misleading to mention that it scales to "thousands" of strategies, when you stop at 1000 (even though technically the statement is accurate).
>
> We admit that our current approach does not scale well for large games. A possible bottleneck is that as we get closer and closer to the equilibrium of the game the number of strategies blows up, resulting in iterations with running time comparable with solving the full LP (see figure 2). A possible workaround, as also mentioned to reviewer yN9W, is to use a fixed number of strategies at each step. Also, we should mention that we performed experiments for a bit larger games (up to 5000 of strategies) and the picture was essentially the same. We just presented the results up to 1000 to cut down on computation time.
>
>  > 2. It seems that it would be good to mention explicitly the existence of a direction that minimizes the directional derivative; of course this follows from the fact that the steepest descent computation can be formulated as an LP over a compact polyhedral set. While it is mentioned in line 204 that the direction can be identified by solving an LP, it seems worth explicitly mentioning this after Theorem 4, before in Lemmas 1 and 3 you make references to a direction that minimizes the ($\rho$)-directional derivative.
>
> We can mention the existence in an updated version.
> > 3. Can you compare to work done using descent methods with the Nikaido-Isoda (NI) function? It seems to be that it might be relevant to mention in related work.
>
> First of all we are thankful to the reviewer for pointing out the NI function. We can compare with this method. We will also add the other references mentioned in Question 1 and expand the literature exposition in the Related Work subsection.
>
> > 4. There should be more information on exactly what the family of block games looks like and how they are generated. It would be good to show running time results for the classes of games that the method does not do well on as well (seems odd to handpick the class for the timing results).
>
> We used block matrices of the following form: we generate a random matrix of size $n/2$ and then pad it with zeros to reach size $n$. In an updated version we can include the explicit definition of the block games, and more running time comparisons, either for the current approach or for the one with fixed number of constraints.

---

### Official Review · Reviewer_S1BM · 2024-11-04

**Soundness:** 2
**Presentation:** 3
**Contribution:** 1
**Rating:** 3
**Confidence:** 4

**Summary:**

This paper studies the problem of approximating the Nash equilibrium in bilinear zero-sum games. In particular, the proposed algorithm applies a steepest descent approach, moving in the direction that minimizes the directional derivative of the duality gap at each timestep. Theoretically, the algorithm achieves an $O(\frac{1}{\rho\delta} log(\frac{1}{\delta}))$ iteration complexity (where $\rho$ is the $\rho$-approximation of the best response query) and converges to a $\delta$-approximate equilibrium. Moreover, the algorithm can be modified via decreasing the schedule to achieve an $O(\frac{1}{\rho} log(\frac{1}{\delta}))$ iteration complexity. Experimentally, the algorithm is shown to require increasing iterations to find an approximate equilibrium as the dimension of the game grows, though the number of iterations needed grows slowly. Moreover, comparisons in running time are made to standard solvers, showing speedups in some specific classes of games.

**Strengths:**

- The paper is well-written and clearly organized. The mathematical exposition is also clearly written.
- The problem studied, namely how to speed up the solving of two-player zero-sum games, is certainly an interesting one at the interface of optimization and game theory.

**Weaknesses:**

- While the running-time benefits of the algorithm seem to be discussed as the key motivation for introducing and analyzing it, the experiments do not seem extensive enough to conclude any beneficial properties of the proposed algorithm. For instance, how do other standard LP-based solver perform from the perspective of the number of iterations in Fig 1? The plots in Figs 2 and 3 also seem fairly arbitrary -- is there a theoretical analysis that can bound the percentage of strategies used as a function of $\delta$ and $\rho$?
- The claims made in the experimental section regarding comparability of performance are also not precise -- what games are the experiments in Table 1 run on? Are they specific to the class of block games described? What other classes of games are there where the algorithms proposed perform better than standard solvers?
- Theorems 1 - 4 are known results/definitional, and thus should not be theorem statements (perhaps leaving them as observations or facts?)
- Overall, while the exposition is nice and the proposed algorithm has its merits, the lack of depth in the analysis and the lack of clear strengths of the algorithm make it difficult to recommend acceptance.

**Questions:**

- In the FIND_DIRECTION algorithm, you re-solve the LP at every time step. Would it make sense to instead use a recursive approach and exploit the convexity of the duality gap to incrementally change $(x', y')$ instead?
- Using learning algorithms with decreasing step-sizes has proven to be useful in the decentralized learning setting. Would such a modification to your algorithm provide any further improvements?

---

> ### Author Response · Authors · 2024-11-21
> **Response to Reviewer S1BM**
>
> We thank the Reviewer S1BM for the review and the questions.
>
> > While the running-time benefits of the algorithm seem to be discussed as the key motivation for introducing and analyzing it, the experiments do not seem extensive enough to conclude any beneficial properties of the proposed algorithm. For instance, how do other standard LP-based solver perform from the perspective of the number of iterations in Fig 1? The plots in Figs 2 and 3 also seem fairly arbitrary -- is there a theoretical analysis that can bound the percentage of strategies used as a function of
> $\delta$ and $\rho$?
>
> In terms of the overall running time, our method does not outperform standard solvers in general 0-sum games, but it does perform favorably for some classes of games. In terms of comparing the number of iterations with other LP-based solvers, this would not be fair; since a comparison only to the number of iterations does not give us insight as to what method is faster, since this also depends on the cost of any iteration. Regarding the bound connecting $\rho$ with the percentage of strategies, no such bound is possible (unless without any further assumptions). To see that, consider matrices with the majority of entries in $(1-\rho, 1]$.
>
> > The claims made in the experimental section regarding comparability of performance are also not precise -- what games are the experiments in Table 1 run on? Are they specific to the class of block games described? What other classes of games are there where the algorithms proposed perform better than standard solvers?
>
> Table 1 is indeed specific to the class of block games. In general the algorithm performs better when there is some “sub-game” structure that leads to a smaller number of LP constraints.
>
> > Theorems 1 - 4 are known results/definitional, and thus should not be theorem statements (perhaps leaving them as observations or facts?)
>
> We can restate them as facts in an updated version (and perhaps move the short proofs to a dedicated appendix to keep the paper self-contained).
>
> > In the FIND_DIRECTION algorithm, you re-solve the LP at every time step. Would it make sense to instead use a recursive approach and exploit the convexity of the duality gap to incrementally change instead?
>
> Actually not resolving a LP in each step would be the biggest step towards achieving significant performance improvements. We proposed a direction in our Conclusions section around warm-starting the LP solver. Does the reviewer have any other proposals on what would be a suitable recursive approach?
> Also, please note that in each step our algorithm uses an exact solver (i.e. scipy’s default tolerance of $10^{-9}$). Since the directional derivative is already an approximation, one could get away with using approximate LP solvers as well; at least for all but the last few iterations.
>
> > Using learning algorithms with decreasing step-sizes has proven to be useful in the decentralized learning setting. Would such a modification to your algorithm provide any further improvements?
>
> By step size we assume the reviewer means the parameter $\varepsilon$. First we would like to point out the following: our proofs establish that there is some $\varepsilon$ such that $V(z + \varepsilon (z’-z)) \le V(z)$ where $z = (x,y)$. Given that $V$ is convex function it follows that $V(z + \varepsilon (z’-z))$ is a convex function of $\varepsilon$ over $[0,1]$, so one could employ a convex formulation to select the optimal $\varepsilon$ in every step. Now, in practice we noticed that for the first few iterations even $\varepsilon = 1$ works and as the number of iterations grows $\varepsilon$ decreases so we employed a form of linear search throughout the iterations. More recent experiments showed that performing a ternary search for the first few iterations is significantly faster.

---

### Official Review · Reviewer_yN9W · 2024-11-04

**Soundness:** 3
**Presentation:** 3
**Contribution:** 2
**Rating:** 3
**Confidence:** 4

**Summary:**

The authors bimatrix zero-sum games and provide a convex approach to provide a gradient-descent algorithm on the duality gap function (as a minmization problem, instead of minmax) and show that their method converges at rate $O(1/\varepsilon\log(1/\varepsilon))$ to a NE of the game.

**Strengths:**

1) The paper is generally well written and the methods are clearly explained.

2) The results are presented in an intuitive manner and the experiments are conducted that show the efficacy of their theoretical results.

**Weaknesses:**

1) The contribution of this paper with respect to the novelty (technically) and the problem they are trying to solve could be better explained.

2) For two-player zero-sum games which is the setting studied here it is well known from the equivalence to Linear Programs that one can obtain $O(poly(size).polylog(1/\varepsilon))$ convergence to the Nash equilibrium, which is polynomial in the size of the representation of the LP.

3) An important point to note in the literature is that the algorithms for which last-iterate convergence is studied are predominantly *no-regret* (online) algorithms, which have numerous consquences even beyond two-player zero-sum games, for instance convergence to CE/CCE's in multiplayer games etc. Hence the challenge is obtain last-iterate for such algorithms, see for example [Golowich et al., 2020].

4) For example a direction that would be interesting (even empirically) is to investigate the time to converge to NE for very large zero-sum games and compare to algorithms such as OGDA, OMWU etc.



References:

Golowich, Noah, Sarath Pattathil, and Constantinos Daskalakis. "Tight last-iterate convergence rates for no-regret learning in multi-player games." Advances in neural information processing systems 33 (2020): 20766-20778.

**Questions:**

Please see weaknesses

---

> ### Author Response · Authors · 2024-11-21
> **Response to Reviewer yN9W**
>
> We thank the Reviewer yN9W for the review and the questions.
>
> We answer each question/weakness below:
> > 1. The contribution of this paper with respect to the novelty (technically) and the problem they are trying to solve could be better explained.
>
> The technical novelty of the paper lies in the fact that we performed descent directly on the function of the duality gap, which also happens to be a convex function (instead of applying the usual gradient descent approaches on each player’s utility function, which is not convex). Most works use either different distances (KL divergence, $l_1$ error etc), or introduce more complex proximity measures to show convergence. We believe that our contribution, given the directional derivative, can potentially lead to other works in this direction.
>
> > 2. For two-player zero-sum games which is the setting studied here it is well known from the equivalence to Linear Programs that one can obtain convergence to the Nash equilibrium, which is polynomial in the size of the representation of the LP.
>
> This is indeed correct. Although so far, such approaches may not be used in practice, despite their theoretical guarantees, which was in part the motivation for our work, i.e., to approach two player zero-sum games and LP with first order methods. This is a point that has already been pointed out in the bibliography (see Gilpin et. al. reference paper). Furthermore, there is a speed up from $O(poly(size) poly (\log(\frac{1}{\varepsilon})))$ to $O(poly(size) \log(\frac{1}{\varepsilon}))$ in our approach.
>
> > 3. An important point to note in the literature is that the algorithms for which last-iterate convergence is studied are predominantly no-regret (online) algorithms, which have numerous consquences even beyond two-player zero-sum games, for instance convergence to CE/CCE's in multiplayer games etc. Hence the challenge is obtain last-iterate for such algorithms, see for example [Golowich et al., 2020].
>
> We agree with the reviewer. While carrying out our work, we did not study whether or not our method is no-regret. We recently tried to obtain results on this, but have not yet managed to have any conclusions. Therefore, we leave this as a very interesting question for future work.
>
> > 4. For example a direction that would be interesting (even empirically) is to investigate the time to converge to NE for very large zero-sum games and compare to algorithms such as OGDA, OMWU etc.
> We tried to compare with OMWU, with parameters as in the Daskalakis and Panageas (ITCS 2019) paper. It turns out that OMWU is cyclic with respect to the duality gap (and the behaviour persists even for smaller $\eta$s).
>
>
> Furthermore, If we want to solely focus on empirical testing there is a modification of our approach that greatly outperforms the current one: instead of approximating the direction using $\rho$-best responses, one can simply choose to always work with k best responses out of the set of pure strategies (either for a constant k of for some function of n). We can prove that this method converges quite easily: for each iteration i, the k best responses belong to some $\rho_i$- best response set hence the total running time can be upper bounded by $\frac{1}{\rho_{max}} log(\frac{1}{\delta})$ where $\rho_{max}$ is the maximum $\rho_i$. While that guarantee is not better than the one presented in the paper, it avoids the blowup of the strategies used as we approach the equilibrium. Would the reviewer be interested in some experiments about this approach?

---

> > ### Comment · Reviewer_yN9W · 2024-11-25
> > **Some additional comments**
> >
> > I thank the authors for their clarifications. I believe, OGDA should perform better in practice, than OMWU, although in theory both algorithms have last-iterate convergence guarantees for bilinear zero-sum games.
> >
> > In my view, what would make the paper stronger are the following:
> >
> > a) Explicitly stating the convergence guarantees and compare it with guarantees for OMWU, OGDA, all in terms of the appropriate game parameters. Note that for instance, OWMU will have a dependence on a game dependent constant, which is pretty much unavoidable! See [Wei et al., 2020], [Cai et al., 2024].
> >
> > b) Perform experiments on large scale zero-sum games and compare both the number of iterations and the wall-clock time for your approach and OMWU, OGDA etc.
> >
> > c) If the algorithm is shown to be no-regret, this will have additional benefits.
> >
> > References:
> >
> > Wei, Chen-Yu, et al. "Linear last-iterate convergence in constrained saddle-point optimization." arXiv preprint arXiv:2006.09517 (2020).
> >
> > Cai, Yang, et al. "Fast Last-Iterate Convergence of Learning in Games Requires Forgetful Algorithms." arXiv preprint arXiv:2406.10631 (2024).

---

### Meta-Review · Area_Chair_Hg9P · 2024-12-20

**Metareview:**

The authors propose a steepest descent type algorithm on the the duality gap function to find approximate Nash equilibrium in two player zero-sum games. The reviewers believe that there is a lack of novelty in the paper and the approach and none of the reviewers was enthusiastic about the paper. The AC also agrees with their opinion and we recommend rejection.

**Additional Comments On Reviewer Discussion:**

After the rebuttal, the reviewers still were not convinced that the paper is above the bar for acceptance.

---

### Decision · Program_Chairs · 2025-01-22

Reject